# The Novel Function of Unsymmetrical Chiral CCN Pincer Nickel Complexes as Chemotherapeutic Agents Targeting Prostate Cancer Cells

**DOI:** 10.3390/molecules27103106

**Published:** 2022-05-12

**Authors:** Jing-Jing Qu, Lin-Lin Shi, Yan-Bing Wang, Jing Yan, Tian Shao, Xin-Qi Hao, Jia-Xiang Wang, Hong-Yu Zhang, Jun-Fang Gong, Bing Song

**Affiliations:** College of Chemistry, School of Life Sciences, School of Basic Medical Science, The First Affiliated Hospital of Zhengzhou University, Zhengzhou University, No. 100 of Science Road, Zhengzhou 450001, China; qujing0621@163.com (J.-J.Q.); 13027781853@163.com (Y.-B.W.); tjnuyj@163.com (J.Y.); shao_tian@zzu.edu.cn (T.S.); xqhao@zzu.edu.cn (X.-Q.H.); wjiaxiang@zzu.edu.cn (J.-X.W.)

**Keywords:** pincer complex, nickel, prostate cancer, androgen receptor, PSA

## Abstract

We report that the pincer nickel complexes display prostate cancer antitumor properties through inhibition of cell proliferation. Notably, they display better antitumor properties than cisplatin. Mechanistic studies reveal that these pincer nickel complexes trigger cell apoptosis, most likely due to cell cycle arrest. Interestingly, these complexes also inhibit androgen receptor (AR) and prostate-specific antigen (PSA) signaling, which are critical for prostate cancer survival and progression. Our study reveals a novel function of pincer nickel complexes as potential therapeutic drugs in prostate cancer.

## 1. Introduction

Prostate cancer (PCa) is the first and most common form of cancer and the second leading cause of cancer-related death among men in the United States. One in five men is expected to be diagnosed with PCa during his lifetime [1]. It is also the most rapidly increasing cancer type among men in China [2]. Although the 5-year survival rate of localized PCa is nearly 100%, metastatic PCa remains a deadly disease. While most advanced types of PCa initially respond to androgen deprivation therapy (ADT) with tumor regression, a majority of them eventually progress to a resistant disease referred to as castration-resistant prostate cancer (CRPC). Recent use of new-generation antiandrogens such as abiraterone and enzalutamide (Enz) further prolonged patient lives for several months to a year; nevertheless, resistance inevitably develops, ultimately leading to death [3,4,5,6]. Sipuleucel-T (Provenge) is the only FDA-approved immunotherapy vaccine made for prostate cancer patients [7]. Balancing cost effectiveness, chemotherapy is still the first treatment used for advanced prostate cancer, although side effects are common. Thus, there is an urgent need to develop novel chemotherapeutic drugs against PCa.

Metal-based complexes have been used in cancer therapy since the discovery of the platinum compound (cisplatin) in the 1960s. They have the advantage of potential chemotherapeutic and cost-effective properties. Cisplatin has been the most successful anticancer drug used to treat many types of cancer; however, it has an uncertain effect on PCa and moderately decreases prostate-specific antigen (PSA), mainly due to its cellular toxicity and acquired resistance [8,9,10,11,12]. For these reasons, anticancer compounds with other kinds of metal are being investigated. Therefore, ruthenium, gold, and iron anticancer complexes have been tested to avoid severe side effects. The working mechanism and toxicity profile of drugs incorporating other metals than platinum may be distinct from platinum drugs and, thus, offer advantages over existing therapeutic regimens [13,14,15,16].

Nickel is abundant in our environment and ubiquitous in a variety of foods. Although in a trace amount, it is essential in physiological processes in our body [17]. Nickel complexes could be applied as valuable alternatives to anticancer platinum agents. Indeed, nickel-incorporated complexes have shown anticancer properties in prostate, breast, lung, and colon cancer cells [18,19,20].

In a previous study, we synthesized and characterized a series of pincer nickel complexes [21]. In the present study, we tested the cell viability of prostate cancer cells after treatment of these organometallic pincer compounds that have been previously designed, synthesized, and characterized by our group. Four pincer nickel complexes showed significant inhibitory effects in prostate cancer cells, triggering cell cycle arrest and higher levels of apoptosis with low IC_50_. This inhibitory effect occurs in both androgen-sensitive and androgen-insensitive prostate cancer cells. The mechanism of reduced cell growth is due to the repression of AR/PSA survival signaling, which is vital in promoting prostate cancer.

## 2. Results

### 2.1. Chemical Synthesis

We previously synthesized a series of pincer metal compounds in order to screen and test the biological function of these compounds. The synthesis of unsymmetrical chiral ligand precursors and the pincer nickel metal compounds is shown in Figure 1 [21].

### 2.2. Cytotoxic Studies

Since these compounds were stable under normal physiological conditions, we assessed the cytotoxic effect of these compounds on two representative prostate cancer cells−LNCaP (androgen-sensitive) and PC-3 (androgen-insensitive) cells. WST-1 test was performed with concentrations of compounds at 0, 1, 2.5, 5, and 10 μM on all cell lines. Four compounds showed significant inhibition of growth in LNCaP cells (Figure 1). It is interesting to note that these four compounds also showed significant inhibition of cell growth in PC-3 cells, which are considered androgen-insensitive and cisplatin-resistant cell lines (Figure 2). In summary, these results indicate that these pincer nickel complexes have growth inhibitory activity toward prostate cancer cells. Using logistic regression, IC_50_ concentrations were determined at 48 h of treatment in LNCaP cells (Table 1). Thus, we chose 5 μM for further mechanism study.

### 2.3. Cell Cycle Analysis

Regulation of the cell cycle is a homeostatic mechanism that maintains proper cellular function. Many chemotherapeutic drugs have demonstrated their anticancer effects through disruption of the cell cycle. To reveal the impact of these compounds on the cell cycle, flow cytometric analysis was performed. LNCaP cells were treated with 5 μM of different compounds for 48 h. As shown in Figure 3, all four compounds increased the proportion of sub-G1 stage cells, which are considered to be apoptotic cells. This is consistent with our previous proliferation assay results. Interestingly, compounds **4****a**, **4b**, and **4d** induced S phase arrest, whereas compound **4c** caused G2/M phase arrest. These results indicate that these pincer nickel compounds are able to induce cell cycle arrest, followed by cell apoptosis.

We performed Annexin V-APC/7-AAD staining analysis to improve our cell death analysis. As shown in Figure 4A, the early and late apoptotic rates were markedly increased in cells treated with compounds **4a** and **4c**, compared with the control. In addition, there was no significant difference in the number of early apoptotic cells among the groups treated with compounds **4a** and **4c**. Notably, there were more late apoptotic cells in the group treated with **4a** (Figure 4B). These results are consistent with our previous cell cycle analysis.

### 2.4. Cell Death Analysis

Cleaved PARP is one of the most used diagnostic markers for the detection of apoptosis. To investigate the effect of the compounds in inducing the intrinsic apoptotic pathway, we examined the levels of cleaved PARP protein (Figure 5). LNCaP cells were treated with 5 μM of compounds **4****a** and **4c** for 48 h and subjected to Western blot analysis. As expected, the expression of apoptotic-protein-cleaved PARP was high in groups treated with compounds **4****a** and **4c**, compared with that in the control group. The activation of this protein led to cell apoptosis, which is consistent with our proliferation and FACS analysis.

### 2.5. AR and PSA Expression Analysis

It is well recognized that AR signaling plays a central role in the development and progression of PCa. Prostate-specific antigen (PSA) expression is the clinically validated downstream indicator of androgen receptor activity and an important marker for prostate cancer progression in patients. Thus, we treated LNCaP cells with chemicals **4a** or **4c** and examined expressions of AR and PSA. Figure 4 shows that AR was highly expressed in LNCaP cells. However, in the presence of chemical **4a** or **4c**, the expression of AR protein significantly decreased. The PSA expression also decreased, which is consistent with our observation of cell proliferation.

## 3. Discussion

One of the main characteristics of PCa is hormonal reactivity. Androgen receptor activation is necessary for the normal growth of the prostate. Aberrant AR signaling is also one of the major reasons for prostate cancer occurrence and progression. Early-stage PCa initially responds to androgen deprivation therapy (ADT) with tumor regression but eventually develops into metastatic castration-resistant prostate cancer (mCRPC) [22,23]. In this study, the pincer nickel complexes were screened for their antiproliferative activity against prostate cancer cells. Interestingly, they showed cytotoxic activity on both androgen-sensitive prostate cancer cells (LNCaP) and androgen-insensitive prostate cancer cells (PC-3). Further mechanistic studies suggest that these pincer nickel complexes kill prostate cancer cells via apoptosis. This is, in part, due to cell cycle arrest and repressed expressions of AR signaling. The downregulation of PSA observed in LNCaP cells treated with complexes is consistent with their ability to downregulate AR. Since the PSA gene is one of the major AR regulated genes, it is speculated that the reduction in AR leads to a decline in PSA.

## 4. Materials and Methods

### 4.1. Chemical Synthesis

Cisplatin (Beyotime Biotechnology, Shanghai, China, S1552) was prepared and used as suggested by the manufacturers. Chemicals were synthesized as previously described [21]. In a 25 mL two-necked Schlenk tube, compound **3** (0.44 mmol), NaOAc (0.36 g, 4.40 mmol), and NiCl_2_ (0.11 g, 0.88 mmol) were added in dry DMAc (10 mL). The mixture was refluxed under an Ar atmosphere for 24 h. After the mixture was cooled and concentrated in vacuo, the residue was purified by passing through a short column containing a layer of Celite and a layer of silica with dichloromethane as eluent. The desired products of **4** (including **4a**, **4b**, **4c**, and **4d**) were purified once again by preparative TLC on silica gel plates with CH_2_Cl_2_/EtOAc 1/1 as eluent.

### 4.2. Cell Culture and Reagents

PCa cell lines LNCaP and PC-3 were obtained from Stem Cell Bank, Chinese Academy of Sciences. All cell lines were authenticated and tested free of mycoplasma. LNCaP and PC-3 cells were cultured in RPMI 1640 with 10% fetal bovine serum and 1% penicillin–streptomycin. The cells were maintained at 37 °C in a humidified incubator with 5% carbon dioxide.

### 4.3. Cell Proliferation Assay

The proliferation assay was performed using WST-1 (Beyotime Biotechnology, Shanghai, China, C0036), as described by the manufacturer. In brief, 3000 LNCaP or PC-3 cells/well were seeded in 96-well plates. Cells were either treated with vehicle (DMSO) or different concentrations of compounds, 10 μL of WST-1 was added to the cultures and incubated for 2 h when harvesting, and the absorbance was measured at 450 nm using a spectrophotometer.

### 4.4. Cell Cycle and Apoptosis Analysis

Cell cycle and apoptosis analysis were performed as described by the manufacturer (Beyotime Biotechnology, Shanghai, China, C1052). Cells were harvested via trypsinization, fixed in 75% ethanol, stained with propidium iodide solution at a final concentration of 50 ug/mL, and subjected to fluorescence-activated cell sorting (FACS) analysis.

### 4.5. Antibodies and Western Blots

Antibodies used in the Western blots were GAPDH (Abcam, Shanghai from China, ab9484), cleaved PARP (Cell Signaling, Shanghai, China. 5625S), androgen receptor (Cell Signaling, Shanghai, China. 5153S), and PSA (Cell Signaling, Shanghai from China, 2475S); all antibodies were diluted as suggested by the manufacturers. Western blot analyses were performed using standard protocols. In brief, cell lysates were dissolved in 1XNP40 sample buffer, sonicated, and quantified by Bradford Protein Assay Kit (Beyotime Biotechnology, Shanghai, China, P0006) and then boiled with 1XSDS loading dye for 10 min at 95 °C, separated on a 4–20% Precast PAGE gel for Tris-Gly system (Beyotime Biotechnology, Shanghai, China, P0468S), and transferred to a PVDF membrane. The membranes were blocked with 5% milk in PBST for 1h at RT, incubated in primary antibody diluted in primary antibody dilution buffer (Beyotime Biotechnology, Shanghai, China, P0023A) overnight at 4 °C, washed 3 × with PBST, and incubated for 1 h in a secondary antibody (Sangon Biotech, Shanghai, China, 1:5000). Membranes were washed 3 × with PBST and subjected to development.

### 4.6. Statistical Analysis

Two-tailed paired *t*-tests were used to assess the statistical significance of cell functional assays.

## 5. Conclusions

Although various ways have been investigated to treat prostate cancer, chemical therapy is still the first line of treatment for prostate cancer [23]. In the present study, we showed that pincer nickel compounds have antiproliferative activity for prostate cancer cells. Androgen insensitivity is one of the major reasons for prostate cancer therapy failure. These types of nickel complexes have inhibitory activity toward both androgen-sensitive and androgen-insensitive cell lines, indicating that pincer nickel complexes can be used in patients with androgen insensitivity.

Nickel has been reported to induce mitochondrial apoptosis and caspase-dependent apoptosis [17]. Whether it is the compound or the nickel ion that plays a major role in the results generated in our study still needs further investigation. We also observed that all four complexes induced cell cycle arrest and cell death, but complex 13 exhibited significant mitotic arrest other than S-phase arrest. This may indicate a different mechanism.

Taken together, we showed that pincer nickel complexes exert antiproliferative activity via induction of cell cycle arrest. These complexes can also downregulate the protein expression of AR and PSA in LNCaP cells. Our research suggests a new category of complexes with anticancer properties. 

## Data Availability

Not applicable.

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
