# Peer review of "The Novel Function of Unsymmetrical Chiral CCN Pincer Nickel Complexes as Chemotherapeutic Agents Targeting Prostate Cancer Cells"

_molecules, 2022, doi:10.3390/molecules27103106_

Round 1
Reviewer 1 Report
The authors in this original paper named:” Novel Function of Unsymmetrical Chiral CCN Pincer Nickel Complexes as Chemotherapeutic Agents Targeting Prostate Cancer Cells” showed a novel applications of nickel complexes to treat prostate cancer cells.
The experiments have been well-conducted. However, this reviewer has several points need to be amended:
1.- Viability experiments: only the authors used SubG1 by flow cytometry to represent the viability the cells in the different days (more specificity 3 days). But to this reviewer the authors must improve the viability experiments to corroborate the death (Caspase, Anexin V experiments) in the Fig 4, as wells as total PARP-1 are missed.
2.- Normal/Control cells: the author only showed the effect of nickel complexes 4a-4c in prostate cancer cells (LNCaP), the author must be improve the effect of these novel components in normal cells, to see the possible adverse effect in normal/control cells.
Author Response
POINT-BY-POINT REBUTTAL
Manuscript ID: molecules-1670153
Type of manuscript: Article
Title: Novel Function of Unsymmetrical Chiral CCN Pincer Nickel Complexes as
Chemotherapeutic Agents Targeting Prostate Cancer Cells
Review comments:
Comments and Suggestions for Authors
The authors in this original paper named:” Novel Function of Unsymmetrical Chiral CCN Pincer Nickel Complexes as Chemotherapeutic Agents Targeting Prostate Cancer Cells” showed a novel applications of nickel complexes to treat prostate cancer cells.
The experiments have been well-conducted. However, this reviewer has several points need to be amended:
1.- Viability experiments: only the authors used SubG1 by flow cytometry to represent the viability the cells in the different days (more specificity 3 days). But to this reviewer the authors must improve the viability experiments to corroborate the death (Caspase, Anexin V experiments) in the Fig 4, as wells as total PARP-1 are missed.
Response: Thank you for your suggestion. We have performed Annexin V-APC/7-AAD staining analysis to improve our cell death analysis. As shown in Figure. 4A, the early and late apoptotic rate were markedly increased in the compound 4a and 4c treated cells compared with in the control. In addition, there was no significant difference in the number of early apoptotic cells among the 4a and 4c treated groups. Notably, there was more late apoptotic cells in the 4a treated group (Figure. 4B). These results are consistent with our previous cell cycle analysis.
A
B
Figure. 4 Cell apoptosis analysis using Annexin V-APC/7-AAD staining and flow cytometry in LNCaP cells treated with complexes. (A) Representative image of cell apoptosis analysis by flow cytometry. Early-stage apoptotic cells are presented in the lower right quadrants (Annexin V-APC positive and 7AAD negative), and late-stage apoptotic cells are presented in the upper right quadrants (Annexin V-APC and 7AAD positive). (B) Quantitative analysis of early, late stage and total apoptotic cell percentage in LNCaP cells treated with compounds. All experiments were repeated three times for data analysis. (*P < 0.05,***P < 0.001)
2.- Normal/Control cells: the author only showed the effect of nickel complexes 4a-4c in prostate cancer cells (LNCaP), the author must be improve the effect of these novel components in normal cells, to see the possible adverse effect in normal/control cells.
Response: Thank you for your suggestion. We have considered using RWPE-1, a normal prostate cell line as our control. However, due to the covid-19 pandemic, we have had difficulty in purchasing the cell and its culture medium.

Reviewer 2 Report
Cancer stands as a major cause of mortality and morbidity worldwide, still being a main subject of research in terms of aetiology and pathogenesis. Numbers and figures coming from the 2022 Cancer’s Facts and Figures report of the American Cancer Society still look concerning: in fact, 1.9 million of total new cases are expected to be diagnosed during 2022 in the US, together with 609,360 expected deaths, which sums up to about 1,670 deaths per day. Continuously performed research during the course of the last decadesmanaged to identify the four major contributors to cancer’sdeath rates and cancer-related morbidity: lung & bronchial cancer, prostate cancer, breast cancer, and colorectal cancer.
The study hereby reviewed aims to open new perspectives for what concerns prostate cancer’s treatment; this form of cancer is known to be the most common malignancy in American men, with 1 in 5 men expected to be diagnosed at some point during their life. Mortality rates reported are significant, with prostate cancer representing the second most lethal malignancy for men right after lung cancer. However, statistics also display a big discrepancy in between death rates for localized and metastatic prostate cancer, with the former having an almost 100% 5-year survival rate compared to the latter, which is a deadly disease. Unfortunately, advanced prostate cancer responds to androgen deprivation therapy (ADT) only for a limited amount of time and just in a partialsubset of cases: this leads to the definition of the entity known as castration-resistant prostate cancer (CRPC), for which chemotherapy still remains the first line of treatment especially in terms of cost-effectiveness, despite the numerous side effects.
Considering the great therapeutic qualities that metal-based chemotherapy complexes showed along the years, with emphasis for those platinum-based, the authors investigatedthe potential role of pincer nickel compounds (synthesized in their own laboratories) in the treatment of prostate cancer, taking into account the individual characteristics of androgen-sensitive and androgen-insensitive forms, and listing their results based on cytotoxic studies, cell cycle analysis, cell death analysis and androgen receptor (AR) with PSA expression analysis.
Parameters used for analysis, reported in the results section, represent well-established markers often mentioned in the related modern literature, and thoroughly described. Tables and charts are visually intuitive, reporting malignant cell’s survival rate crossed with inhibitory substance’sconcentration, and readily communicate the main body of information that constitutes the rationale of the study, while a deeper and more detailed outlook is offered in the corresponding paragraph for results. Methods and materials are meticulously portrayed with detailed mentioning of sources and protocols related to cell cultures (LNCaP and PC-3) and reagents (4a, 4b, 4c, 4d), standards of preparation for the therapeutic agents under study, and type of molecular assays implemented for efficacy detection, including specific antibodies detected in western blots (GAPDH, cleaved PARP, Androgen Receptor, PSA). Two-tailed paired t-tests were used to assess statistical significance of the datas obtained.
The main, novel discovery of the present study is that pincer nickel compounds exhibit anti-proliferative activity against prostate cancer cells, regardless of their profile for androgen hormones sensitivity, which represent to this day the most common cause for therapeutic failure against advanced prostate cancer. Given this aspect, it becomes clear how potential implications in terms of cancer morbidity and mortality reduction on a global scale are substantial. Compounds 4a, 4b, 4d induced S phase arrest whereas compound 4c caused G2/M phase arrest, indicating that these pincer nickel compounds are able to induce cell cycle arrest,followed by cell apoptosis. It is worth mentioning how the authors couldn’t unequivocally prove whether the nickel ion itself or the compound form play the biggest role in apoptosis induction, therefore further research is definitely needed to elucidate the real impact of the present study. Lastly, funding sources and conflicts of interest are listed in the final paragraph in order to raise the level of transparency for this study.
Author Response

(The authors gave the same response as above.)

Round 2
Reviewer 1 Report
No answer given.